# Implicit Neural Representations for Deformable Image Registration

**Jelmer M. Wolterink**                                J.M.WOLTERINK@UTWENTE.NL
**Jesse C. Zwienenberg**                  J.C.ZWIENENBERG@STUDENT.UTWENTE.NL
**Christoph Brune**                                           C.BRUNE@UTWENTE.NL
*Department of Applied Mathematics & Technical Medical Centre,*
*University of Twente, Enschede, The Netherlands*

## Abstract

Deformable medical image registration has in past years been revolutionized by the use of convolutional neural networks. These methods surpass conventional image registration techniques in speed but not in accuracy. Here, we present an alternative approach to leveraging neural networks for image registration. Instead of using a convolutional neural network to *predict* the transformation between images, we optimize a multi-layer perceptron to *represent* this transformation function. Using recent insights from differentiable rendering, we show how such an implicit deformable image registration (IDIR) model can be naturally combined with regularization terms based on standard automatic differentiation techniques. We demonstrate the effectiveness of this model on 4D chest CT registration in the DIR-LAB data set and find that a three-layer multi-layer perceptron with periodic activation functions outperforms all published deep learning-based results on this problem, without any folding and without the need for training data. The model is implemented using standard deep learning libraries and flexible enough to be extended to include different losses, regularizers, and optimization schemes.

**Keywords:** Image registration, neural networks, implicit neural representations, chest CT, regularization

## 1. Introduction

With the advent of deep learning in medical image analysis, convolutional neural networks (CNNs) have been widely applied to deformable image registration (Fu et al., 2020). A common approach is to use training data to optimize a CNN which, given two new and unseen images, predicts a deformation vector field (DVF) on a grid (de Vos et al., 2019; Dalca et al., 2018; Eppenhof et al., 2018). During training, reference DVFs are available (Eppenhof et al., 2018) or obtained indirectly through optimization of an image similarity metric (de Vos et al., 2019; Balakrishnan et al., 2019). Extensions to this approach include, e.g., multi-stage (Hering et al., 2019, 2021) or adversarial (Elmahdy et al., 2019) training. Once trained, these methods are generally faster than *conventional* iterative approaches (Sotiras et al., 2013), but at the cost of reduced accuracy (Hansen and Heinrich, 2021b). Moreover, learning-based require large training sets. To address this last issue, several works have aimed to combine deep learning with iterative optimization at inference by optimizing a CNN for each new image pair (Laves et al., 2019; Fechter and Baltas, 2020).

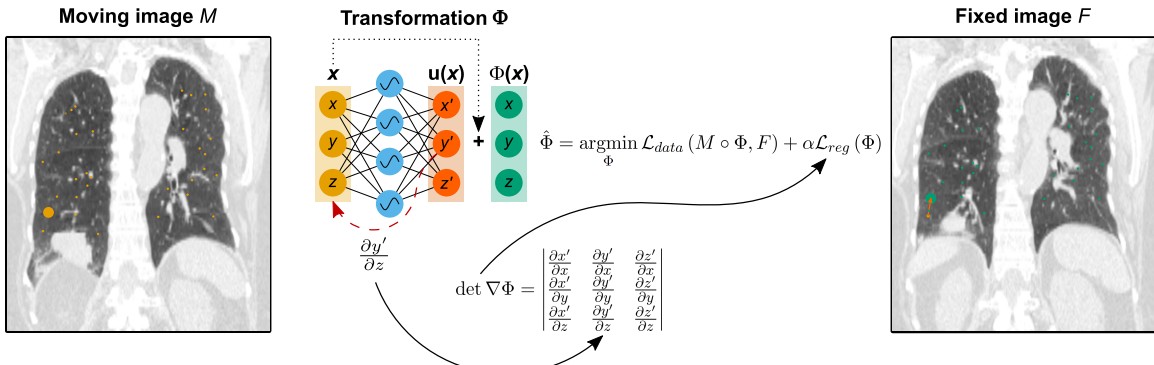

Figure 1: We optimize a multi-layer perceptron (MLP) that *implicitly* represents the function $\Phi$ mapping coordinates from a moving to a fixed image. The MLP is optimized using gradient descent with a similarity metric and a regularization term, computed using analytical gradients of the MLP. Periodic activation functions (Sitzmann et al., 2020) allow the balancing of small and large deformations.

CNNs are widely used for image registration, but typically as an operator that maps between an image and a DVF function. Here, we propose to directly optimize the DVF as a function on a spatial domain. We build on recent advances in differentiable rendering and optimize implicit neural representations, in which a function on a spatial domain is *implicitly* represented in the weights of an MLP that operates on (continuous) coordinates in that domain (Tancik et al., 2020; Mildenhall et al., 2020; Park et al., 2019). For each new image pair, we optimize a network that takes as input any spatial coordinate and provide as output a function value. In the case of deformable image registration, we are looking for a function $\Phi(\mathbf{x}) = u(\mathbf{x}) + \mathbf{x}$ that maps each location $\mathbf{x}$ in one image to a location in a second image (Fig. 1).

Parametrizing the function $\Phi$ as an INR in a multi-layer perceptron has several advantages for image registration. First, the network represents a function that is continuous or *meshless* and not restricted to any particular grid resolution. This allows using the same lightweight model regardless of image resolution, while a CNN architecture would have to be adapted for different image sizes. Second, we can exploit functionality in deep learning libraries such as TensorFlow and PyTorch to numerically compute gradients of the transformation with respect to coordinates. This leads to more accurate gradients than finite difference approximations (Balakrishnan et al., 2019; de Vos et al., 2019) and allows us to draw from a vast body of literature on efficient regularizers for medical imaging, e.g. (Burger et al., 2013; Rueckert et al., 1999). Third, we can modify the activation function used in the network (Sitzmann et al., 2020) and therewith modify the neural tangent kernel of the network (Tancik et al., 2020), i.e., its ability to represent (local) large or small deformations.

In this work, we propose an algorithm for intensity-based implicit deformable image registration (IDIR) that has sufficient flexibility to include higher-order regularization terms. We demonstrate its feasibility on 3D deformable image registration in the DIR-LAB 4D chest CT data set.

## 2. Methods

We consider pairwise image registration, in which the goal is to find an optimal spatial transformation between images $F$ and $M$. Both images are described in the same domain, i.e., $M : \Omega \subset [-1,1]^n \to \mathbb{R}, F : \Omega \subset [-1,1]^n \to \mathbb{R}$. Our objective is to find a transformation $\Phi(\mathbf{x}) = u(\mathbf{x}) + \mathbf{x}$ on the image coordinates such that coordinate $\mathbf{x}$ in image $F$ anatomically corresponds to coordinate $\Phi(x)$ in image $M$, i.e., $(M \circ \Phi)(\mathbf{x}) = F(\mathbf{x}) \; \forall \mathbf{x} \in \Omega$.

Finding this transformation can be posed as an optimization problem

$$\hat{\Phi} = \underset{\Phi}{\operatorname{argmin}} \, \mathcal{L}_{data}\left(M \circ \Phi, F\right) + \alpha \mathcal{L}_{reg}\left(\Phi\right), \tag{1}$$

where $\mathcal{L}_{data}$ is a similarity metric between the fixed image $F$ and the deformed moving image $M$, $\mathcal{L}_{reg}$ is a regularization term on the transformation $\Phi$, and $\alpha$ is a weighing term. The key insight of our work is that we represent the transformation $\Phi$ as a neural network (Fig. 1). In contrast to CNN-based image registration methods (Dalca et al., 2018; de Vos et al., 2019; Hering et al., 2021; Fechter and Baltas, 2020), this network does not take image intensities as inputs, but a coordinate $\mathbf{x} \in \Omega$. As output, it returns a new coordinate $\Phi(\mathbf{x}) = u(\mathbf{x}) + \mathbf{x}$. Hence, the network parameters defining the transformation can be optimized using standard stochastic gradient descent methods based on an image similarity loss. The network is a lightweight multi-layer perceptron, whose input $\mathbf{x}$ is a (continuous) coordinate from the image domain, $u(\mathbf{x})$ is a deformation vector predicted by the neural network, and the addition required to obtain $\Phi(\mathbf{x})$ is simply a residual connection.

### 2.1. Implicit neural representation

By the universal approximation theorem (Hornik et al., 1989), any well-behaved function can be described with arbitrarily high accuracy by an appropriate neural network. In our application, the network may vary in depth or width, but always has $n$ input nodes and $n$ output nodes for transformations in $n$-dimensional Euclidean space. Hence,

$$\Phi(\mathbf{x}) = u(\mathbf{x}) + \mathbf{x} = \mathbf{W}_i \left(\phi_{i-1} \circ \phi_{i-2} \circ \ldots \circ \phi_0\right)(\mathbf{x}) + \mathbf{b}_i + \mathbf{x}, \; \mathbf{x}_i \mapsto \phi_i\left(\mathbf{x}_i\right) = \sigma\left(\mathbf{W}_i \mathbf{x}_i + \mathbf{b}_i\right). \tag{2}$$

Here, $\phi_i : \mathbb{R}^{M_i} \mapsto \mathbb{R}^{N_i}$ is the $i$-th layer in the network, $\mathbf{W}_i \in \mathbb{R}^{N_i \times M_i}$ is a weight matrix, $\mathbf{x}_i \in \mathbb{R}^{M_i}$ is an input vector, and $\mathbf{b}_i \in \mathbb{R}^{N_i}$ is a trainable bias. A standard choice for the element-wise activation function $\sigma$ is the commonly used rectified linear unit (ReLU) $\sigma(x) = \max(0, x)$. However, ReLUs have a bias towards low-frequency signals (Mildenhall et al., 2020; Tancik et al., 2020). This means that the model might find it difficult to represent small local deformations in image registration.

Two common approaches to overcome the low-frequency bias of ReLU activation functions are to pre-process the input coordinates with periodic activation functions (Fourier feature mapping) (Mildenhall et al., 2020; Tancik et al., 2020) or to replace the ReLU activation function with a periodic activation function (Sitzmann et al., 2020). In this work, we choose the latter, and for $\sigma$ choose a periodic activation function to obtain a SIREN model, i.e. $\sigma(x) = \sin(x)$. An added benefit of periodic activation functions in SIREN networks is that they can be differentiated multiple times. This substantially expands the set of regularization terms that can be used in the network, as we will demonstrate in Sec. 3.

## 2.2. Regularization

Because deformable image registration is an ill-posed problem, it is common practice to regularize the DVF to avoid unrealistic deformations. CNN-based registration methods represent DVFs as samples on a voxel grid, and thus can only approximate spatial gradients with finite-difference schemes (Balakrishnan et al., 2019; de Vos et al., 2019). For example, a partial derivative $\frac{\partial x'}{\partial x}$ in the Jacobian matrix in Fig. 1 would be approximated as $\frac{\partial x'}{\partial x} \approx u(x + \delta x, y, z) - u(x, y, z)$, where $\delta x$ is equal to the voxel size, which can be in the order of millimetres. This leads to numerical errors that might in turn affect registration (Fig. 3).

In IDIR, all operations in the neural network (Eq. 2) are differentiable, and we can instead *analytically* compute gradients of the output deformation vector with respect to the input coordinates using modern deep learning libraries like PyTorch or Tensorflow. For example, we can easily fill the Jacobian matrix row-wise by taking the analytical gradients of the network output w.r.t. each coordinate $(x, y, z)$. This matrix can be computed for any point in the image domain $\Omega$, not just for points on a voxel grid. While the network using a ReLU activation function is differentiable once, the network using a periodic activation function is differentiable more than once. Therefore, we can compute any number of common regularization terms and include them in the optimization of the network. We here illustrate this concept with three regularization terms that can be added as $\mathcal{L}_{reg}$ to Eq. 1.

**Jacobian regularizer** The Jacobian determinant $\det \nabla \Phi$ at location $\mathbf{x}$ is an indicator of local expansion and shrinkage, where a negative determinant indicates folding and the loss of invertibility. While shrinkage and expansion should occur locally, we want to limit large deviations from 1.

$$\mathcal{S}^{\mathrm{jac}}[\Phi] = \int_\Omega |1 - \det \nabla \Phi| d\mathbf{x}.$$

**Hyperelastic regularizer** Additional constraints on the DVF can be imposed in a hyperelastic regularization term (Burger et al., 2013). It consists of a length, surface area, and volume term to control variations in all of these aspects. The length is controlled by the Jacobian matrix of the deformation $u$. The cofactor matrix and the determinant of the Jacobian matrix of the transformation control the area and volume respectively. The function $\phi_{\mathrm{c}}$ penalizes the expansion of area, and $\psi$ is a convex function that satisfies $\psi(v) = \psi(1/v)$ and hence penalizes growth and shrinkage equally.

$$\mathcal{S}^{\mathrm{hyper}}[\Phi] = \int_\Omega \left[ \frac{1}{2}\alpha_{\mathrm{l}}|\nabla u|^2 + \alpha_{\mathrm{a}}\phi_{\mathrm{c}}(\mathrm{cof}\,\nabla\Phi) + \alpha_{\mathrm{v}}\psi(\det \nabla\Phi) \right] d\mathbf{x},$$

with the convex functions: $\phi_{\mathrm{c}}(C) = \sum_{i=1}^3 \max\left\{ \sum_{j=1}^3 C_{ji}^2 - 1, 0 \right\}^2$ and $\psi(v) = \frac{(v-1)^4}{v^2}$.

**Bending energy penalty** Smoothness of the deformation vector field can be further imposed using the bending energy penalty proposed in (Rueckert et al., 1999). This requires that the second derivatives of the deformation are small everywhere in the domain.

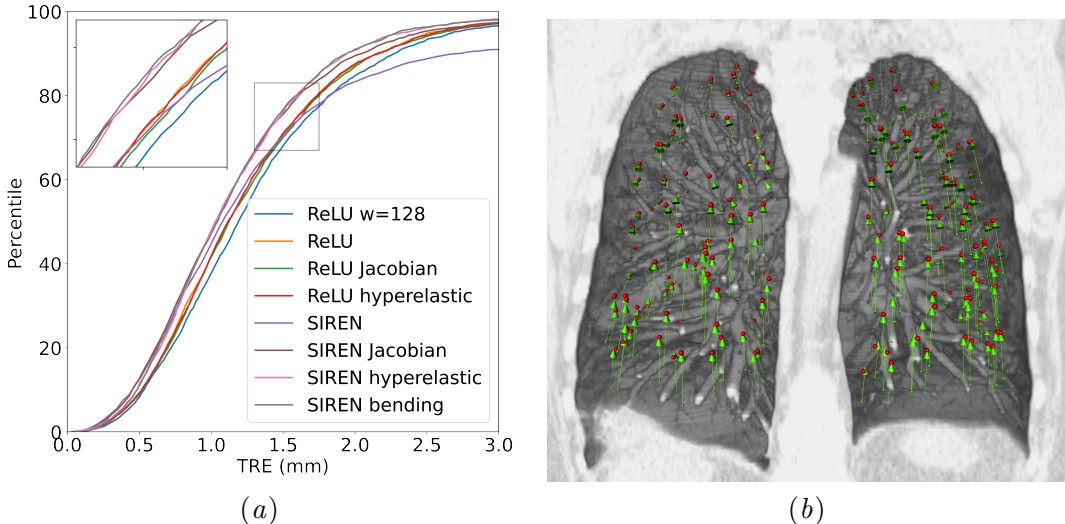

Figure 2: (a) Cumulative distribution of TRE in DIR-LAB set for different settings of our IDIR model. (b) Example deformation vector field obtained on landmarks in DIR-LAB 4DCT 08, with target landmarks in red and deformation vectors in green.

$$\mathcal{S}^{\text{bending}}[\Phi] = \frac{1}{8} \int_{-1}^{1} \int_{-1}^{1} \int_{-1}^{1} \left[ \left( \frac{\partial^2 \Phi}{\partial x^2} \right)^2 + \left( \frac{\partial^2 \Phi}{\partial y^2} \right)^2 + \left( \frac{\partial^2 \Phi}{\partial z^2} \right)^2 \right. \\ \left. +2 \left( \frac{\partial^2 \Phi}{\partial xy} \right)^2 + 2 \left( \frac{\partial^2 \Phi}{\partial xz} \right)^2 + 2 \left( \frac{\partial^2 \Phi}{\partial yz} \right)^2 \right] dxdydz \tag{3}$$

For a network that uses ReLU activation functions, it would be pointless to compute this term: The second derivative of a ReLU is 0, and thus this term is 0 everywhere. In contrast, in a SIREN network, the second derivative can be computed as the derivative of a periodic function, and we can include the regularizer.

## 3. Experiments and Results

To evaluate the potential of our method, we perform 3D deformable image registration in the DIR-LAB data set (Castillo et al., 2009). In this set of 4D CT images of ten patients, the task is to register inspiration images to expiration images. This is a challenging problem due to superposition of cardiac and respiratory motion that substantially exceed the scale of the small lung structures. The data set has been widely used to evaluate image registration methods, including deep learning-based ones, e.g. (de Vos et al., 2019; Eppenhof et al., 2018; Hering et al., 2021). As is common practice (Rühaak et al., 2017), we only optimize the transformation for the points within a lung mask obtained in the inspiration image using a deep learning algorithm (Hofmanninger et al., 2020). Images have varying resolution,

Table 1: Mean (standard deviation) DIR-LAB target registration error (TRE) in mm of the proposed IDIR framework, [1]a state-of-the-art iterative algorithm using isotropic total variation regularization (Vishnevskiy et al., 2017), [2]a CNN-based iterative algorithm (Fechter and Baltas, 2020), [3]the current state-of-the-art in deep learning DIR (Hering et al., 2021), [4]the TRE before registration (Castillo et al., 2009).

| Scan | IDIR (ours) | isoPTV[1] | CNN[2] | VIRNet[3] | Displacement[4] |
|---|---|---|---|---|---|
| 4DCT 01 | 0.76 (0.94) | 0.76 (0.90) | 1.21 (0.88) | 0.99 (0.47) | 4.01 (2.91) |
| 4DCT 02 | 0.76 (0.94) | 0.77 (0.89) | 1.13 (0.65) | 0.98 (0.46) | 4.65 (4.09) |
| 4DCT 03 | 0.94 (1.02) | 0.90 (1.05) | 1.32 (0.82) | 1.11 (0.61) | 6.73 (4.21) |
| 4DCT 04 | 1.32 (1.27) | 1.24 (1.29) | 1.84 (1.76) | 1.37 (1.03) | 9.42 (4.81) |
| 4DCT 05 | 1.23 (1.47) | 1.12 (1.44) | 1.80 (1.60) | 1.32 (1.36) | 7.10 (5.14) |
| 4DCT 06 | 1.09 (1.03) | 0.85 (0.89) | 2.30 (3.78) | 1.15 (1.12) | 11.10 (6.98) |
| 4DCT 07 | 1.12 (1.00) | 0.80 (1.28) | 1.91 (1.65) | 1.05 (0.81) | 11.59 (7.87) |
| 4DCT 08 | 1.21 (1.29) | 1.34 (1.93) | 3.47 (5.00) | 1.22 (1.44) | 15.16 (9.11) |
| 4DCT 09 | 1.22 (0.95) | 0.92 (0.94) | 1.47 (0.85) | 1.11 (0.66) | 7.82 (3.99) |
| 4DCT 10 | 1.01 (1.05) | 0.82 (0.89) | 1.79 (2.24) | 1.05 (0.72) | 7.63 (6.54) |
| Average | 1.07 | 0.95 | 1.83 | 1.14 | 8.52 |

ranging from $256 \times 256$ pixel to $512 \times 512$ pixel in-plane resolution. Because our INR is resolution independent, we use the same network and hyperparameters for each image.

For each image pair, a model is optimized for 2500 epochs. In each epoch, 10,000 points are randomly sampled from the masked image domain. Networks are optimized with Adam, with a learning rate of $10^{-4}$. The IDIR framework allows for any differentiable metric to be used as loss term $\mathcal{L}_{data}$, but we here use the normalized cross-correlation between sampled intensities in the fixed image and corresponding intensities in the moving image. The algorithm is implemented in Python using PyTorch[1]. Gradients are computed using the PyTorch `autograd` functionality. The second derivatives in Eq. 3 are computed by taking the gradient of the Jacobian matrix elements with respect to the input variables. Sub-voxel image intensity values are computed through trilinear interpolation. The total time required to register two 3D image volumes increases with the number of backward passes per epoch, namely 15 sec (no regularization, 1 pass/epoch), 1 min (Jacobian regularization, 4 passes/epoch), 2 min (hyperelastic regularization, 5 passes/epoch), or 5 min (bending energy penalty, 13 passes/epoch) on an NVIDIA RTX 2080 Ti GPU.

## 3.1. Quantitative Evaluation

As in other works, we evaluate in the DIR-LAB 4D CT dataset by computing the target registration error (TRE) for 300 predefined anatomical landmarks per CT scan pair (Castillo et al., 2009). Figure 2(a) shows a cumulative distribution of TRE values across all 3,000 landmark points in the data set. In this visualization, adapted from (Rühaak et al., 2017; Hering et al., 2021), lines that are shifted towards the left represent higher accuracy.

---

1. Code is available on https://github.com/MIAGroupUT/IDIR

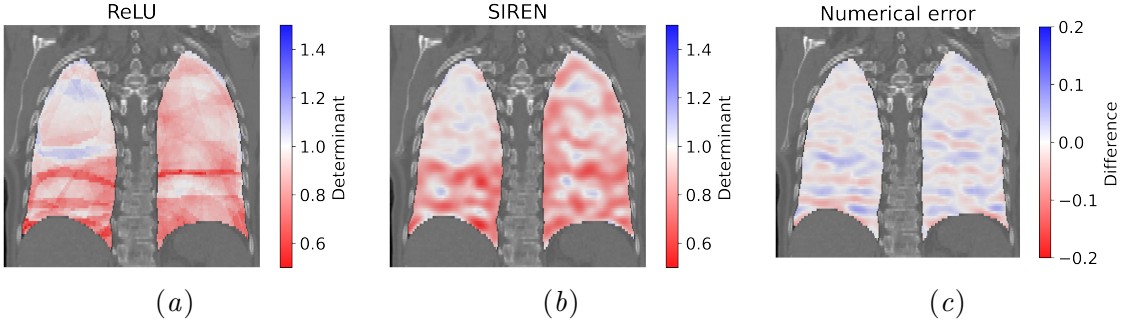

Figure 3: Jacobian matrix determinant at each point in the lung mask, for networks with (a) a ReLU activation function and (b) a periodic activation function. (c) Difference between numerically approximated and analytically computed determinants.

First, a comparison between a ReLU model with 3 layers consisting of 128 hidden units and one with 256 hidden units shows that adding units leads to an increase in performance. Second, we compare ReLU and SIREN activation functions in the 3 layer, 256 hidden units network. The SIREN model without regularization results in several large errors in DIR-LAB 4DCT 08 and 09. This is likely due to the tendency of the network to fit high-frequency signals, or small local deformations. However, with each of the three proposed regularization terms, the SIREN network outperforms the standard ReLU model by a margin, while regularization barely has any effect on the ReLU model. All three generalization strategies perform similarly in the SIREN network. Figure 2(b) visualizes deformation vectors predicted on the landmarks in DIR-LAB 4DCT 08, showing that the model captures both small and large deformations.

Table 1 lists DIR-LAB results obtained with SIREN activation functions and a bending energy penalty (IDIR), a state-of-the-art iterative approach (Vishnevskiy et al., 2017), a CNN-based one-shot registration approach (Fechter and Baltas, 2020), the most accurate published deep learning approach evaluated on this problem (Hering et al., 2021), and the displacement prior to registration. These results show that our multi-layer perceptron outperforms existing deep learning methods, which often include multiple stages, pre-alignment, and require large training data sets with annotations. Moreover, the model outperforms one-shot learning with a CNN. Conversely, TRE values are still slightly higher than those reported by the best conventional iterative registration method.

Figure 3 visualizes the determinants of the Jacobian matrix resulting from a network trained with ReLU activation functions and a SIREN network. The low-frequency ReLU activation function results in a composition of coarse piecewise linear deformations, while the high-frequency SIREN activation function results in more localized deformations. In both these cases, as in all registration results of our method included in Table 1, all determinant values were non-negative. Hence, we did not observe any folding in any of the image pairs, i.e., $\nabla\Phi > 0$ for all voxels in the lungs. Figure 3(c) shows the difference between Jacobian determinants of the SIREN model obtained with analytical gradients, and those obtained using numerical approximation with a finite difference scheme as in CNN-based methods.

## 4. Discussion and Conclusion

We have presented IDIR, a novel approach to deformable medical image registration that uses a multi-layer perceptron to implicitly represent a transformation function. We regularize the optimization of this function by analytically computing gradients with respect to any location in the image domain. This allows us to compute, e.g., the determinant of the Jacobian matrix directly, without the use of finite differences. We have demonstrated the feasibility of this approach to 4D CT registration in the DIR-LAB data set.

Our work shows how a DVF between images can be optimized and implicitly represented in a lightweight neural network that outperforms all published deep learning registration methods on the DIR-LAB data set. While we use a multi-layer perceptron as implicit neural representation, previous works (Laves et al., 2019; Fechter and Baltas, 2020) have aimed to integrate CNNs in iterative registration. A direct comparison on DIR-LAB shows that our work might have benefit over using a CNN to output a discrete voxel-based DVF (Fechter and Baltas, 2020). Part of this may be due to improved precision in the continuous function that we optimize, and the ability to compute accurate analytical gradients.

We found that the activation function is an important design choice that affects registration. A network using ReLU activation functions was able to provide reasonable registration results due to its bias to low-frequency functions, which is suitable for chest CT registration. However, explicit regularization barely had effect on the ReLU network. This might be because there is a lower limit to the scale of deformations that the ReLU network can fit. The bias of neural networks with ReLU activation functions towards low-frequency functions (Mildenhall et al., 2020) can be addressed with Fourier feature mappings Tancik et al. (2020) or periodic activation functions (Sitzmann et al., 2020). Our SIREN experiments showed that periodic activation functions allowed the network to represent high frequency signals, and thus small local deformations. For image registration in general, being able to balance between small and large deformations is a desirable property.

We have here demonstrated a basic single-stage version of IDIR on a benchmark registration data set with a standard normalized cross-correlation loss. Exciting future research directions remain, as the model is sufficiently flexible that it can be combined with other differentiable losses, regularization terms, and multi-stage approaches, and data sets. Moreover, our implementation could be further accelerated using recent advances in implicit neural representations (Müller et al., 2022) and frameworks like JAX. While we found the Jacobian determinant to be non-negative in our transformations – indicating diffeomorphic registration – the model, and thus the transformation, is not directly invertible. In future work, this might be overcome by using invertible networks (Jacobsen et al., 2018).

In contrast to deep learning-based registration methods, IDIR does not require large sets of training data, and instead optimizes a new network for each image pair. However, there are ways to embed IDIR as a deformation model in a learning framework. For one, MLPs could be pretrained on population data so that they only require fine-tuning at inference. Moreover, the MLP could be conditioned on latent vectors that describe the moving and fixed image. Finally, it would be interesting to investigate regularization of the DVF on sparsely sampled keypoints using, e.g., graph networks (Hansen and Heinrich, 2021a).

In conclusion, neural networks are a feasible approach to implicitly representing a continuous transformation function for deformable medical image registration.

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
