# OpenReview forum: "Implicit Neural Representations for Deformable Image Registration"
_MIDL.io/2022/Conference — MIDL 2022_

### Official Review · Reviewer_xsdN · 2022-01-23

**Confidence:** 5
**Preliminary Rating:** 4
**Recommendation:** Oral, Poster

**Summary:**

This paper presents IDIR, a deformable registration method for medical images that iteratively optimizes a neural network as implicit representation of the deformation field per image pair. The authors build their work on recent advances in neural rendering and propse an approach to learn a continuous function that outputs the displacement for a given voxel coordinate of the input image.

By adopting the recent SIREN activation function instead of ReLUs, the network can be differentiated multiple times, which allows for straightforward implementation of regularizers that make use of higher-order derivatives. The authors further use three different regularizers to impose smoothness constraints on the displacement field.

The method is experimentally evaluated on 4D CT images from the DIR-LAB data set and is compared against different state-of-the-art methods. While the method does not outperform all comparisons, it does outperform recent deep learning based methods without the need for any prior training. Using SIREN activations and proper regularization, the determinants of the Jacobians suggest a diffeomorphic registration.

**Strengths:**

First of all, I really enjoyed reading this paper. It is well-written and easy to follow, although presenting a sophisticated method. The presented approach is interesting and a logical application of recent advances in neural rendering. The method is explained clearly and its beauty lies in its simplicity. Moreover, the method performs very well and does not need prior training on large data sets. The use of higher-order derivatives is especially interesting in the context of regularizers in deformable registration, which is exploited by the proposed method. The experimental evaluation highlights the potential of the method. I applaud the authors for boldly making a comparison with a method (isoPTV) that outperforms the approach presented. This helps a lot to judge the capabilities of the method. I appreciate the (anticipated) publication of the code.

**Weaknesses:**

While the application of implicit neural representations to registration is of high interest, the approach is not as radically novel as the authors claim. Laves et al. (2019) have already applied the very similar concept of Deep Image Prior to deformable registration and there exist some works that have used similar approaches for, e.g., CT reconstruction, inpainting, or denoising of medical images (Baguer et al., 2020; Tölle et al., 2021). I admit that there are some differences between Deep Image Prior and the coordinate-based interpolation used in the presented paper. However, the relevant prior work should properly be cited. Additionally, the presented method comprises very little novelty compared to the seminal works from neural rendering. I still think that the work contains enough novelty to be published, but the authors should clearly state their contribution.

The method is only evaluated on a single task and data set. The experimental evaluation could be more elaborate. Especially since the method does not require prior training. Other than that, the paper is valid and sound.

Laves, M. H., Ihler, S., & Ortmaier, T., 2019. Deformable medical image registration using a randomly-initialized CNN as regularization prior. MIDL 2019. arXiv:1908.00788.

Baguer, D.O., Leuschner, J., Schmidt, M., 2020. Computed tomography recon- struction using deep image prior and learned reconstruction methods. Inverse Problems 36, 094004.

Tölle, M., Laves, M. H., & Schlaefer, A., 2021. A Mean-Field Variational Inference Approach to Deep Image Prior for Inverse Problems in Medical Imaging. MIDL 2021.

**Deanonymize Review:**

no

**Detailed Comments:**

Some minor comments:

* Abstract: "a single three-layer multi-layer perceptron" should be rephrased.
* I don't understand the use of $ \omega $ and the weight initialization in § 2.1. This could be explained a bit more detailed or removed and refered to Sitzmann et al. (2020).
* I would appreciate a sentence about the intuition behind bending energy as it is done for the other two regularizers.

**Final Rating After The Rebuttal:**

4: Weak Accept

**Justification Of The Final Rating:**

The authors responded well to my questions and I think this work is a good contribution to MIDL. The combination of deep learning methods and classical optimization offers great potential, since large labeled datasets are usually difficult to obtain. However, given the limited methodological novelty, my original assessment remains.

**Paper Type:**

both

**Questions To Address In The Rebuttal:**

* Consideration of relevant prior work as mentioned above.
* Clear contribution statement.

Otherwise, the paper is fairly clear and raises no questions that would need to be addressed in the rebuttal.

**Special Issue:**

no

---

### Official Review · Reviewer_Svam · 2022-01-24

**Confidence:** 5
**Preliminary Rating:** 5
**Recommendation:** Oral

**Summary:**

This is an exiting work that explores the possibility of rephrasing iterative medical image registration as an instance optimisation over neural network parameters that can represent a spatial transformation. The method can employ any mathematical regulariser without crude fixed-point approximations and the authors further extend the idea to higher order derivatives by using a periodic activation (SIREN). That way smoother deformations can be obtained with Jacobian-determinant or hyper-elastic regularisation. The approach is validated on the DIRlab 4DCT dataset is very accurate and has a reasonably runtime at inference (but it is still much slower than current DL-based or conventional algorithms).

**Strengths:**

- The concept is novel, convincingly motivated and a very nice contribution to MIDL
- The accuracy (and likely the deformation smoothness, zero negative Jacobian determinants are reported) is excellent outperforming most (if not all) DL-based registration approaches
- The paper is very well written, easy to follow and nicely illustrated.
- The method could be further extended using other cost functions (e.g. semantic features) and will be mode open source.

**Weaknesses:**

- I am missing some better intuition why a simple random sampling was performed. Lung images exhibit prominent geometric features (e.g. bifurcations) that could substantially reduce the required samples (cf Ruehaak).
- It would be good to discuss potential extensions w.r.t. to the network architecture. While it is impressive that a basic fully-connected network works well and can be realised with a reasonable number of trainable parameters, it could also be conceivable that a GraphCNN (e.g. Deep learning based geometric registration for medical images IPMI 2021) could work well in this scenario (where a sparse number of coordinate points is sampled). Currently the runtimes are pretty low (although that is usually not a problem).
- Some details about the implementation, in particular the metric, normalised cross correlation, are missing. What window size was employed?
- A discussion of the relations to one-shot learning is missing: see e.g. "One Shot Learning for Deformable Medical Image Registration and Periodic Motion Tracking" T Fechter, TMI 2020

**Deanonymize Review:**

yes

**Final Rating After The Rebuttal:**

5: Strong Accept

**Justification Of The Final Rating:**

as mentioned in my post-rebuttal comment, the authors further improved their work and there isn't a higher rating than strong accept ;-) I also continue to recommend this paper for a special issue extension

**Paper Type:**

methodological development

**Questions To Address In The Rebuttal:**

I would like to get some additional information of how the twice differentiable step is implemented in pytorch. It would also be good to get some empirical evidence that finite differences (as commonly used) do really deteriorate the quality of e.g. a hyper-elastic regulariser.
For further discussion, I would think it could be interesting to explore population pre-training of the model, as some aspects that drive the neural network for deformation representation could be shared across scan pairs.

**Special Issue:**

yes

---

### Official Review · Reviewer_AYgp · 2022-01-27

**Confidence:** 4
**Preliminary Rating:** 3
**Recommendation:** Poster

**Summary:**

The authors apply and evaluate implicit neural representations (INRs) in the context of deformable image registration: instead of training a neural network to predict deformation fields between pairs of images, the network is used to represent a transform. The work analyzes the effect of choosing periodic (SIREN) activation functions over ReLU and explores the benefit of different regularization terms. Accuracy in terms of target registration error (TRE) is computed for the DIR-LAB dataset consisting of lung-CT scans from 10 subjects, and compared to results reported by other studies, where it outperforms deep-learning (DL) methods.

**Strengths:**

1. The work evaluates INRs in the context of image registration, using only standard DL tools such as the PyTorch library and stochastic gradient descent, and demonstrates the use of INRs as a means of parameterizing deformable transforms.

2. The authors show that INRs can accurately represent transformations between lung-CT images from the same subject. Their approach achieves registration accuracy in terms of TRE surpassing the performance of DL methods reported in the literature.

3. The paper reproduces and confirms the superiority of SIREN over ReLU activation for representing dense transforms, previously demonstrated by Sitzmann et al. for images, wavefields, video, sound, and their derivatives.

**Weaknesses:**

1. The comparison with DL methods, with the stated disadvantage of requiring “large training data sets”, appears inadequate. The appeal of DL methods is the ability to train on one set of images to later register unseen images from a different set, whereas INRs need to be learned separately for every new image pair. The DL-typical GPU-enabled libraries and stochastic gradient descent could be used in the same way to optimize a voxel-wise displacement field instead of an INR. As an alternative parameterization, INRs are not in competition with DL registration. Rather than spinning the work as a stand-alone registration method, it would be preferable to show if, and how, INRs can be as accurate as competing parameterizations - for an identical optimization strategy.

2. Parameterizing transforms with INRs is technically interesting, but some of the advantages presented in the introduction are not convincing. First, the benefit of image-size independence is unclear as the network needs to be retrained for every new image pair. Second, it is unclear how the ability to use a range of regularizers and to represent various deformation scales differentiates INRs from alternative parameterizations.

3. The evaluation is limited to images from 10 subjects and only lists TRE achieved by competing methods in the respective publications.

**Deanonymize Review:**

no

**Detailed Comments:**

The authors report that increasing network size improves performance, which is well known (see e.g. [1]). It would be interesting to analyze what capacity is needed to match the accuracy of competing parameterizations (likely not in the rebuttal).

[1] Hoffmann et al. SynthMorph: learning contrast-invariant registration without acquired images. IEEE Transactions on Medical Imaging (2021).

**Final Rating After The Rebuttal:**

4: Weak Accept

**Justification Of The Final Rating:**

I would like to thank the authors for their detailed response, which addresses my comments well. The revision has improved the quality of the manuscript and made the work more focused, and I have therefore increased my review score. For the final version, I would recommend highlighting the lowest TRE value in each row of Table 1 in bold font.

**Paper Type:**

validation/application paper

**Questions To Address In The Rebuttal:**

The background on regularization terms appears excessive and is well known, as shown by the references provided by the authors. Some of the space could be better used to address weaknesses 1-2.

It is unclear why it is desirable to avoid finite difference approximations, and how this could practically be achieved when parameterizing deformations using INRs as part of a DL method.

The cumulative distribution of TRE in Figure 2 is difficult to interpret, e.g. between ReLU with 128 vs. 256 hidden nodes. A box plot might be a better choice.

It is unclear what expiration and inspiration images are. Why is it relevant that DIR-LAB is widely used?

The authors should stress right at the beginning that retraining is required for every new image pair. In this regard, the method practically is a classical approach using a DL parameterization.

The gradient computation is presented as the most time-consuming part of the optimization. Can this be quantified?

TRE is computed for 300 landmarks per CT scan. How were these chosen?

The benefit of regularization has also been studied (see e.g. [1]). However, it would be interesting to know why it had no effect on the ReLU INR.

It is unclear what “proper regularization” means, and how the regularization weights were chosen for the regularizers tested.

The authors report that large errors for the SIREN INR without regularization are “likely due to the tendency [...] to fit small local deformations”. Can this be confirmed quantitatively?

The “most accurate published deep-learning approach” is a strong statement. This should be made specific. Do the authors refer to the best results published for the specific dataset? It should be clearly stated early on that the authors run only their implementation, and compare to TRE values reported elsewhere.

Without consulting the reference, it is unclear what “error by experts” pertains to. Likewise, it is unclear what “local composition of large deformations” for the ReLU INR means, and how this is determined.

**Special Issue:**

no

---

### Meta-Review · Area_Chair_K2vS · 2022-02-19

**Recommendation:** Accept (Oral)
**Confidence:** 5

**Metareview:**

All reviewers and myself appreciate the interesting idea here in combining the INR representation with registration, and appreciate the cleanliness of the paper and execution of the idea. The thorough discussion has also indeed improved the paper -- I appreciate the authors's thorough response and willingness to build on the feedback -- which is what we want to see from these discussions.

I would encourage the authors to explore the MLP-vs-DIP aspect a bit more in the future. I'm not really convinced that either is the optimal representation for a deformation field, and might even vary by task. It would be an interesting analysis and paper on its own.

Congratulations to the authors.

---

### Decision · Program_Chairs · 2022-02-28

Accept